# Cell Culture-Derived Tilapia Lake Virus-Inactivated Vaccine Containing Montanide Adjuvant Provides High Protection against Viral Challenge for Tilapia

**DOI:** 10.3390/vaccines9020086

**Published:** 2021-01-25

**Authors:** Weiwei Zeng, Yingying Wang, Huzi Hu, Qing Wang, Sven M. Bergmann, Yahui Wang, Bo Li, Yuefeng Lv, Hua Li, Jiyuan Yin, Yingying Li

**Affiliations:** 1Guangdong Provincial Key Laboratory of Animal Molecular Design and Precise Breeding, School of Life Science and Engineering, Foshan University, Foshan 528000, China; okhuali@fosu.edu.cn (H.L.); yinjiyuan@prfri.ac.cn (J.Y.); 2Key Laboratory of Aquatic Animal Immune Technology, Key Laboratory of Fishery Drug Development, Pearl River Fisheries Research Institute, Chinese Academy of Fishery Sciences, Ministry of Agriculture, Guangzhou 510380, China; eagles5257@prfri.ac.cn (Y.W.); huhuzi@liyang-tech.com (H.H.); 1904028156@stu.tjau.edu.cn (Y.W.); m190100122@st.shou.edu.cn (B.L.); 1904028142@stu.tjau.edu.cn (Y.L.); liyingying@prfri.ac.cn (Y.L.); 3Institute of Infectology, Friedrich-Loffler-Institut (FLI), Federal Research Institute for Animal Health, Greifswald-Insel Riems, 17493 Greifswald, Germany; Sven.Bergmann@fli.de

**Keywords:** inactivated vaccine, tilapia lake virus, adjuvant, protection

## Abstract

Tilapia lake virus (TiLV) is a newly emerging pathogen responsible for high mortality and economic losses in the global tilapia industry. Currently, no antiviral therapy or vaccines are available for the control of this disease. The goal of the present study was to evaluate the immunological effects and protective efficacy of formaldehyde- and β-propiolactone-inactivated vaccines against TiLV in the presence and absence of the Montanide IMS 1312 VG adjuvant in tilapia. We found that β-propiolactone inactivation of viral particles generated a vaccine with a higher protection efficacy against virus challenge than did formaldehyde. The relative percent survivals of vaccinated fish at doses of 10^8^, 10^7^, and 10^6^ 50% tissue culture infectious dose (TCID_50_)/mL were 42.9%, 28.5%, and 14.3% in the absence of the adjuvant and 85.7%, 64.3%, and 32.1% in its presence, respectively. The vaccine generated specific IgM and neutralizing antibodies against TiLV at 3 weeks following immunization that were significantly increased after a second booster immunization. The steady state mRNA levels of the genes tumor necrosis factor-α (TNF-α), interleukin-1β (IL-1β), interferon γ (IFN-γ), cluster of differentiation 4 (CD4), major histocompatibility complex (MHC)-Ia, and MHC-II were all increased and indicated successful immune stimulation against TiLV. The vaccine also significantly lowered the viral loads and resulted in significant increases in survival, indicating that the vaccine may also inhibit viral proliferation as well as stimulate a protective antibody response. The β-propiolactone-inactivated TiLV vaccine coupled with the adjuvant Montanide IMS 1312 VG and booster immunizations can provide a high level of protection from virus challenge in tilapia.

## 1. Introduction 

Tilapia provides an inexpensive source of protein for the majority of developing countries and comprises the second most important freshwater fish cultured worldwide, with a global production of 6.4 million tons in 2015 [1]. Tilapia is one of the most widely farmed aquaculture species in the world, is relatively easy to propagate because of its tolerance to handling and environmental changes, and exhibits rapid growth with uncomplicated dietary requirements. Although cultured tilapia have historically been unhampered as non-susceptible by disease [2,3,4,5], unregulated cultivation has now led to frequent outbreaks of diseases which are threatening the sustainable growth of tilapia production. Bacterial diseases are becoming more prevalent in farmed tilapia populations [6] and include *Streptococcus* spp. [7,8], *Aeromonas* spp. [9,10] and *Flavobacterium columnare* [11]. In contrast, viral infections were not common in tilapia until recently when a new virus tilapia lake virus (TiLV) emerged, which represents the first major epidemic reported in tilapia [12]. This places the global tilapia industry in jeopardy due to the lethality of the virus [13]. 

TiLV is an enveloped, negative-sense, single-stranded RNA virus with 10 segments encoding 14 proteins [14,15,16] and a diameter between 55 and 100 nm [12,15,17,18]. Viral particles have been found to be sensitive to organic solvents (ether and chloroform) due to their lipid membrane [12]. TiLV was preliminarily identified as an orthomyxo-like virus due to the similarities in the structure of its segment termini [14] and now represents a single new species known as *Tilapia Tilapinevirus* in the new genus Tilapinevirus [19]. TiLV infects tilapia at all growth stages including fertilized eggs, egg yolk larvae, fry, and fingerlings [20,21,22,23]. TiLV is an important risk for the fast-growing tilapia aquaculture worldwide. Currently, outbreaks of tilapia lake virus disease (TiLVD) have been officially declared in 16 countries in Asia, Africa, and South and North America [15,18,24,25,26,27] involving 6 of the 12 major tilapia-producing countries [13,28,29]. Historical samples and export records from tilapia hatcheries in Thailand dating back to 2012 were found to contain the virus, and an advisory was issued to 40 countries including China that are at high risk of TiLV infection due to the import of potentially infected materials from these hatcheries [30]. TiLV is a major threat for Nile tilapia [12,25,31,32] as well as other farmed tilapias including red tilapia (*Oreochromis* spp.) [23], Mozambique tilapia (*Oreochromis mossambicus*) [33], and the hybrids *O. niloticus* × *Oreochromis aureus* [12,15,33] and *O. niloticus* × *O. mossambicus* [34]. TiLV infections have now been detected in wild tilapia in Israel, Malaysia, Peru, and Lake Victoria in Tanzania and Uganda [28]. High morbidity and mortality of TiLV-infected tilapia have been reported from field outbreaks and laboratory-challenged animals [12,23,30], and outbreaks on farms and in wild tilapia lead to mortalities between 20% and 90% [12,15,17,20,30]. Importantly, high mortality rates are characteristic of TiLV outbreaks. However, the total infection rate is presumably much higher, as there are also reports of subclinical infections based on the detection of TiLV nucleic acids [25,35]. 

In experimental trials, TiLV infections of tilapia are associated with a high mortality of 70–90%. TiLV can be transmitted horizontally by cohabitation with infected fish [12,23,33,36] and possibly vertically from broodstock to progeny [37]. In addition, the virus can also occur as a co-infection with other pathogenic microorganisms increasing the severity of TiLV infection [15,26,38]. In May 2017, the World Organization for Animal Health (OIE), the United Nations Food and Agriculture Organization (FAO), the Asia-Pacific network of aquaculture centers (NACA), and the consultative group on international agricultural research (CGIAR) issued announcements and warnings regarding TiLV, urging all members to strengthen quarantine measures to prevent the spread of this virus [20,39,40,41].

To our knowledge, there are no cures for any viral diseases in aquaculture. Vaccination is the most effective way to prevent viral disease, and many kinds of vaccines have been proven successful in reducing the severity of fish viral diseases [42,43,44,45]. Based on the vaccine preparation method, there are four main types of vaccines that were developed against viral infections. These are killed vaccines, attenuated vaccines, subunit vaccines, and DNA vaccines. All four types of vaccine are already commercially available such as the attenuated live vaccine against hemorrhagic disease of grass carp (HDGC) [46] and against cyprinid herpesvirus 3 (CyHV3) [47]; the subunit vaccines against infectious pancreatic necrosis (IPN), infectious salmon anemia (ISA), and salmon rickettsia syndrome (SRS) [47]; and the DNA vaccine against infectious hematopoietic necrosis virus (IHNV) [48]. The majority of viral vaccines commercially available for use in aquaculture are inactivated vaccines [44,46,49,50]. To date, no vaccine or treatment options against TiLV infection in cultured fish are available. Inactivated vaccines would be a good choice against the emerging disease because of their stability, safety, and ease of preparation [44,51]. In the present study, we evaluated the immunological effects and protective efficacy of formaldehyde- and β-propiolactone (BPL)-inactivated vaccines against TiLV to provide a scientific basis for the development of an inactivated vaccine for the prevention of TiLV infections.

## 2. Materials and Methods

### 2.1. Experimental Fish, Cell Lines, and Virus

Nile tilapia (*Oreochromis niloticus* L.) used in infection experiments were kindly provided by a local fish farm in Guangzhou (Guangdong, China). The fish had mean total length and body weight of 12.0 ± 0.5 cm and 20.0 ± 0.5 g, respectively. Possible TiLV contamination in fish and feed were checked by semi-nested PCR to verify the absence of different pathogens [21]. Animal welfare measures were in compliance with the guidelines of the Animal Experiment Committee, South China Agricultural University. Fish were anaesthetized with a 0.1 mL/L mixture of clove oil (Sigma-Aldrich, St. Louis, MO, USA) and 70% ethanol before sacrifice for sample collection. The protocol was approved by the China Guangdong Province Science and Technology Department (permit number: SYXK (Yue) 2014-0136). The tilapia were fed with commercial feed for 14 days before the start of the experiments (Tong Wei, Guangzhou, China). 

The TiLV 2017A isolate that was used to prepare the inactivated vaccines in this study was kindly provided by the Institute of Infectology, Friedrich-Loeffler-Institut Insel Riems in Germany. The tilapia brain (TiB) cell line was used for TiLV propagation [52] and was maintained at 28 °C in M199 medium (Gibco, Grand Island, NY, USA) containing 10% (*v*/*v*) fetal bovine serum (FBS HyClone, GE Healthcare Life Sciences, Logan, UT, USA). FBS concentration in the cell culture medium was reduced to 4% for virus propagation.

### 2.2. Virus Cultivation

TiB cells were inoculated into 150 cm^2^ tissue culture flasks (Corning Inc., Corning, NY, USA) with a confluence of 70–80% and were incubated for 1.5 h at a multiplicity of infection (MOI) of 0.1 of the TiLV 2017A. Afterwards, the medium was replaced by M199 medium containing 4% FBS and incubated at 28 °C. TiLV-infected cell monolayers were harvested at 7 dpi (days post infection) by scraping into M199 and then by centrifuging at 1500× *g* and 4 °C for 10 min. The determination of viral titers with 50% tissue culture infectious dose (TCID_50_) assays was as described previously [42]. Briefly, 100 µL of 10-fold serial dilutions of a TiLV cell culture supernatant was inoculated in duplicate onto TiB cell monolayers in 24-well plates and were incubated for 5 days at 28 °C in a 5% CO_2_ incubator. Cytopathic effects (CPE) were observed by light microscopy, and the TCID_50_ was calculated as previously described [42].

### 2.3. Virus Inactivation

Formaldehyde and β-propiolactone (BPL) (Sigma-Aldrich, St. Louis, MO, USA) were used for virus inactivation and were diluted (*v*/*v*) with PBS to achieve final concentrations of 1:2000, 1:1000, and 1:500 for formaldehyde and 1:3000, 1:2000, and 1:1000 for BPL when added to TiLV preparations. Individual solutions were then incubated at 37 °C or 4 °C for predetermined time intervals, and the infectivity titers and immunogenicity were determined as described in our previous study (Table 1) [42]. Briefly, samples were diluted with M199 medium and inoculated onto TiB cell monolayers in 96-well plates (100 μL/well × 6 wells). After incubation at 37 °C for 1 h, 100 μL of M199 containing 4% FBS was added to each well. CPE and the TCID_50_ were recorded daily for 7 days as described above. To confirm complete inactivation, the supernatants collected at a specific time after inactivation were transferred to fresh TiB cell cultures in 25 cm^2^ flasks for two additional passages. Tilapia (20 fish per group) were administered at 1 × 10^8^ TCID_50/mL_ (before inactivation) of the inactivated virus preparation in a volume of 200 μL by intramuscular injection. Booster immunizations were performed using the same inoculation route and the same dose three weeks later. The negative control group was injected with 200 μL PBS. For 3-week post-booster immunization, all tilapia were challenged with 200 μL TiLV 2017A at a dosage of 1 × 10^7^ TCID_50_/mL via intraperitoneal injection. Fish mortality for each group was recorded for 15 days and used to calculate the survival rate (Table 1).

### 2.4. Preparation of Inactivated Vaccine and Tilapia Immunization

The inactivated virus preparation deemed to be optimal as determined above was added to either equal volumes of PBS or Montanide IMS 1312 VG (SEPPIC, Paris, France) following the recommendations of SEPPIC. The prepared inactivated vaccine was stored at 4 °C and shaken well before use.

Healthy tilapia (*n* = 400) were randomly divided into 8 groups (50 fish per group) that included NC, the negative control group injected with sterile PBS; A, the adjuvant control group injected with sterile PBS mixed with the adjuvant; and groups injected with inactivated vaccine: V1, 1 × 10^8^ TCID_50_/mL; V2, 1 × 10^7^ TCID_50_/mL; V3, 1 × 10^6^ TCID_50_/mL; VA1, 1 × 10^8^ TCID_50_/mL and the adjuvant; VA2, 1 × 10^7^ TCID_50_/mL and the adjuvant; and VA3, 1 × 10^6^ TCID_50_/mL and the adjuvant. Tilapia were injected intramuscularly with inactivated vaccine or PBS and the adjuvant at a volume of 200 μL. Booster immunizations were performed at the same dosages three weeks later.

### 2.5. Sample Collection

Samples were collected pre-vaccination (PV) and at 3 and 6 weeks post primary immunization (WPPI). Blood, spleen, and head kidney were obtained from 5 fish per group at each time point. Clove oil was added to water to anesthetize the fish by immersion before dissection. Sample collection was performed according to previous methods [42]. Briefly, blood was extracted from the caudal veins using a 1 mL sterile syringe and allowed to stand at room temperature for 1–2 h and then stored at 4 °C overnight. The samples were centrifuged at 4 °C at 4000 rpm for 20 min to collect serum and stored at −20 °C until use. Spleen and kidneys were stored in the presence of 200 μL Nucleoprotect RNA (Takara, Otsu, Japan) and then stored at −80 °C.

### 2.6. Measurement of Antibody Response by ELISA

Sera from immunized fish were examined by enzyme linked immunosorbent assay (ELISA) as described previously [53]. Briefly, transparent 96-well plates were coated with 100 μL/well of S8 recombinant protein (1.0 μg/mL) and incubated overnight at 37 °C. The plates were washed 3× with PBST (0.01 M PBS, pH 7.2, and 0.05% Tween 20) and 100 μL/well of 5% (*w*/*v*) skim milk powder diluted in PBS was added. The plates were then incubated at 37 °C for 1 h. Serum diluted 1:200 in PBS (100 μL) was added to each well, and the plates were incubated at 28 °C for 2 h and then washed 5× with PBST. Mouse anti-tilapia IgM monoclonal antibody (100 μL) was added (1:1000) to each well, and the plates were incubated for 1 h at 37 °C. The plates were then washed 5× with PBST, and 100 μL of goat anti-mouse IgG HRP (1:5000; Sigma) was added to each well. The plates were incubated at 37 °C for 1 h and washed 5× with PBST. Tetramethylbiphenyl (100 μL) substrate solution (Proteintech, Manchester, UK) was added, and the plates were incubated for 10 min at room temperature. The reaction was stopped by adding 100 μL of 2 M H_2_SO_4_ to each well, and absorbance at 450 nm was measured using a microplate reader (Tecan, Mannedorf, Switzerland).

### 2.7. Serum Neutralization Test (SNT) 

SNTs were performed as previously described [54]. All serum samples were filtered and heat-inactivated for 30 min at 42 °C before testing. First, 200 μL of tilapia serum samples were diluted 4-fold with M199, and then, 2-fold dilution series were prepared with M199 supplemented with 0.15% bovine serum albumin (BSA), resulting in 1/4 to 1/512 dilutions, and 50 μL diluted sera or M199 supplemented with 0.15% BSA was added to 96-well plates (6 wells for one dilution). Next, 50 μL of TiLV-2017A (1 × 10^5^ TCID_50_/mL) was added to each well of the plate and then incubated at 28 °C for 3 h. After incubation, 100 μL of TiB cell suspension in M199 supplemented with 0.15% BSA (2.0 × 10^5^ to 5.0 × 10^5^ cells) was added to each well of the plates that were then incubated at 28 °C for 7 days. The neutralizing antibody titer of the serum against TiLV was determined as the 50% endpoint of the serum that inhibited CPE in inoculated cells.

### 2.8. Determination of Immune-Related Genes Expression by RT-qPCR

RNA was extracted from spleen and kidney tissues that had been stored at −80 °C (see above) using Nuclelzol according to the manufacturer’s instructions (Takara), and cDNA was prepared as described previously [42]. The gene expression were analyzed with specific primers (Table 2) by real-time quantitative PCR (qPCR) in an ABI PRISM 7500 instrument (Applied Biosystems) using SYBR Premix Ex Taq II kit (Takara). Amplifications were performed in 25 µL final volume using the following cycling conditions: 95 °C for 10 min and 40 cycles of 95 °C for 30 s, 60 °C for 60 s, and 72 °C for 30 s. All qPCR reactions were performed in three biological replicates, and the expression level was corrected by the β-actin content in each sample and expressed as ΔΔCt [55]. All data are presented as relative quantities of mRNA and expressed as the means plus standard errors from three separate experiments. 

### 2.9. Protection against Lethal Challenge and Determination of Viral Load

At 3 weeks following booster immunizations, fish (*n* = 30 for each group) were challenged with 0.2 mL of 1 × 10^6^ TCID_50_/mL TiLV A2017 by intraperitoneal injection. A heating rod in each tank kept the water temperature constant at 28 °C. Clinical signs, morbidity, and mortality were monitored daily. TiLV target tissues including spleen, kidney, and liver were collected from dead fish daily until 16 days post-challenge, when no fish died in all groups. All fish were then euthanized, and the remaining tissues were collected at day 18 post-challenge. The cumulative survival rate was recorded to assess the protection efficacy of the inactivated vaccine, and the relative percent survival (RPS) was calculated according to amends method [59]. The formula was as follows: RPS = (1 − (% mortality of immunized group/% mortality of control group)) × 100. 

A virus-specific Taqman probe was used to determine the viral loads using qRT-PCR in the five tissues of 5 fish in each group to confirm that the fish died of TiLV infection and to determine the antiviral effect of different vaccine and strategies. The commercial One Step PrimeScriptRT-PCR kit (Takara) was used for RT-qPCR with a standard input of 100 ng (5 μL of 20 ng/μL) of the total RNA per reaction. The template RNA was denatured at 95 °C for 5 min prior and utilized the following oligonucleotides targeting TiLV genome segment 1 (S1) (S1-For: 5′-TGACTCAGGAGTTATGCCATT-3′, S1-Rev:5′ -TCCTCATTCCTC GTGGTGTAAGT-3′ and S1-probe, FAM-TATGTTATCTGGTGCTGTTGACT-TAMRA) using the following conditions: 10 μL 2 × RT-PCR buffer III, 0.4 μL Ex Taq HS (5 U/μL), 0.4 μL PrimeScript RT Enzyme Mix II, and 0.4 μL ROX reference dye II 50×), 600 nM each of S1-For and S1-Rev primers, 200 nM of S1-probe, 5 μL RNA, and sterile water to a final volume of 20 μL. The cycling conditions were 42 °C for 30 min and 94 °C for 5 min, followed by 40 cycles of 94 °C/15 s and 59 °C/30 s in ABI 7500 realtime PCR instrument (Applied Biosystems, Foster City, CA, USA). All samples were run in duplicate. A standard curve was generated using 10-fold dilutions of purified standard virus RNA (range 10^0^–10^8^ copies/μL) and were linear within this range (y = 2.99x + 36.754, R^2^ = 0.999) (unpublished data).

### 2.10. Statistical Analysis

All statistical analyses were performed using the SPSS 21.0 package (IBM, Chicago, IL, USA). Differences in the antibody and mRNA expression levels between vaccinated and control fish were analyzed by the Student’s *t*-test. Differences in mortality and RPS were analyzed by one-way ANOVA with the LSD post hoc test. Graphs were generated with GraphPad Prism 5, and trial data are presented as the mean ± standard deviations (SD). In all cases, the differences were considered significant at *p* < 0.05 and extremely significant at *p* < 0.01.

## 3. Results

### 3.1. Virus Inactivation

The TiLV preparations were inactivated with different concentrations of BPL or formaldehyde, and pre-inactivation titers were 10^8.5^ TCID_50_/mL. Treatment with each dilution of the two inactivating agents resulted in a total loss of TiLV infectivity and 1:2000, 1:1000, and 1:500 dilutions of formaldehyde abolished virus infectivity within 60, 36, and 24 h at 37° C, respectively. BPL treatments at 1:3000, 1:2000, and 1:1000 dilutions at 4 °C abolished activity within 60, 36, and 24 h, respectively. TiLV vaccines were prepared using the 1:2000 BPL treatment at 4 °C for 36 h since this gave the highest protection rates (Table 1). 

### 3.2. Detection of Serum IgM and Neutralizing Antibody 

We assessed the levels of anti-TiLV IgM levels pre- and post-vaccination to determine the efficiency of vaccination protocols. Anti-TiLV IgM was not detected in any of the fish pre-vaccination, whereas post-vaccination titers increased in all fish (*p* < 0.05). The titers reached high levels of significance for groups V1, VA1, and VA2 at 3 WPPI (*p* < 0.01), especially in the adjuvant groups. The antigen dose, adjuvant, and booster immunization promoted these titer increases. The control groups A and NC lacked any specific anti-TiLV IgM (Figure 1). These results suggested that our vaccine preparation induced a protective IgM antibody titer. In support of this, tilapia immunized with inactivated virus at high antigen levels (group V1) or in the presence of adjuvant (groups VA1 and VA2) developed detectable neutralizing antibody titers (*p* < 0.05 and *p* < 0.01) at 3 WPPI (Figure 2). A further increase in neutralizing titers was observed after booster immunization at 6 WPPI. All immunized groups developed detectable neutralizing antibody titers with the exception of V3. The neutralizing titers for the VA1 and VA2 groups were significantly higher than those in other inactivated vaccine-immunized groups (*p* < 0.05 and *p* < 0.01, respectively). No antibody response was observed in control fish pre-vaccination or when injected with either PBS or adjuvant alone (Figure 2). 

### 3.3. Immune-Related Gene Expression

We further examined the levels of immune-related gene expression in the spleens and kidneys of fish from the previous experiments at 3 and 6 WPPI. The levels of the CD4, IL-1β, and TNF-α genes in both organs were significantly increased only in groups VA1 and VA2 at 3 WPPI (*p* < 0.05), while IFN-γ and MHC-Ia levels increased significantly in groups V1, VA1, and VA2 in one or both organs. When compared with the PBS and adjuvant-only groups, all the immune-related genes we measured reached the levels of significance for all vaccine groups with the exception of V3 at 6 WPPI in both spleen and kidney tissues (*p* < 0.05 and *p* < 0.01) (Figure 3). These data also indicated that the combination of adjuvant and inactivated vaccine resulted in significantly higher expression levels of immune-related genes and cytokines than the corresponding vaccine without adjuvant. Additionally, these results indicated that humoral (CD4) and cellular immunity (MHC-1ɑ and MHC-II) as well as cytokines (TNF-α, IL-1β, and IFN-γ) were all involved in generating the specific anti-TiLV response in these vaccinated fish.

### 3.4. Determination of Viral Load in Different Tissues after Challenge with TiLV

The goal of an efficacious viral vaccine is a reduction in viral load in the immunized fish. We used a specific Taqman assay using cDNA templates to quantify virus levels in liver, spleen, and kidney tissues of the immunized and unimmunized fish after challenge with virus strain TiLV 2017A. We found that the immunized fish possessed lower levels of TiLV than the control unimmunized fish regardless of the organ we screened. The kidneys showed the highest viral loads among all tissues in the unimmunized fish. In contrast, viral loads were greatly reduced or at the level of detection for liver, kidney, and spleens of the vaccine-immunized fish, especially for groups VA1, VA2, and V1 (Figure 4). These data indicated that the anti-TiLV vaccine greatly reduced the amount of virus.

### 3.5. Protective Effectiveness of the Inactivated Vaccine against TiLV Challenge

All the fish were challenged with a lethal dose of TiLV 2017A at 6 WPPI for evaluation of the immune protection effect of inactivated vaccine. The clinical symptoms and number of dead fish were observed and recorded daily for 16 days post-challenge. We found that the fish began to die on day 5 post-challenge in the 2 control groups, while the first fish to succumb in the vaccine groups occurred one or two days later. These challenge data indicated greater differences in protective effects in the vaccinated fish. The highest RPS we found was 85.7% for VA1, the lowest was 14.3% for V3, and the other groups fell within this range (Figure 5 and Table 3). The samples collected from the livers, spleens, and kidneys of moribund fish in each group during and at the end of the experiment were subjected to a virological examination, where TiLV 2017A used for the challenge was re-isolated and confirmed (data not shown). The dead tilapia developed typical symptoms of TiLV infection including lethargy, loss of appetite, ocular alterations, skin erosions, and abdominal swelling. In conclusion, with the exception of V3, the survival rates for all inactivated vaccine groups were significantly increased against TiLV infection compared with controls (*p* < 0.05). No significant survival differences were observed between the PBS and adjuvant-only controls, demonstrating that the inactivated vaccine could protect the tilapia from TiLV.

## 4. Discussion

Vaccination can be one of the most sustainable ways to prevent losses due to devastating diseases in aquaculture [60]. In this study, the immune responses and protective efficacy elicited by inactivated vaccines against TiLV in the presence and absence of the Montanide IMS 1312 VG adjuvant in tilapia was evaluated. Inactivated vaccines are often preferred for safety reasons and possess the same antigenic composition as infectious virions [61,62]. However, chemical modification of the virus surface does occur in the inactivation process, so the retention of essential viral epitopes is essential [63]. Formaldehyde and BPL have been used as standard agents to prepare inactivated vaccines for humans and animals for many years [64], and we found that both reagents completely eliminated infectivity. However, BPL is more expensive, although its use at 1:2000 displayed the best effect, indicating that it more effectively preserved TiLV immunogenicity over formaldehyde [42,51,64,65]. Additionally, BPL is completely hydrolyzed into nontoxic degradation products [66].

Our trials using live fish indicated that, beginning at 3 weeks following inoculation, high levels of TiLV-specific neutralizing antibodies were induced. This was especially apparent in the 1 × 10^8^ TCID_50/mL_ and 1 × 10^7^ TCID_50/mL_ plus adjuvant trials. Specific antibody titers including neutralizing antibodies usually correspond with the degree of protection seen after a vaccination challenge [42,49,67]. We observed a similar phenomenon in the use of the VA1 and VA2 protocols that generated elevated titers and higher survival rates than the other groups (*p* < 0.05 and *p* < 0.01). This indicated that the elevated specific antibody levels might serve to protect the fish from TiLV infection.

Immune-related gene expression is a surrogate marker of protective immunity and has been widely adopted in fish vaccine studies [42,61,67,68,69]. Six immune response genes were elevated at 3 and 6 weeks following inoculation in this study; we found significant increases in the expression of cytokine genes encoding TNF-α, IL-1β, and IFN-γ and of the CD4 and MHC adaptive immune genes MHC-Ⅰɑ and MHC-II in the spleens and kidneys of most of the vaccinated tilapia. In particular, the use of the adjuvant coupled with booster vaccinations induced a corresponding increase in immune response and protection. Cytokines are a group of cell-signaling molecules that act as a bridge linking the innate and adaptive immune systems [70]. Inflammatory cytokines are represented by lymphokines, interleukins, tumor necrosis factor proteins, and chemokines, depending on their function [70,71]. IL-1β and TNF-α are pro-inflammatory indicators of vaccine efficacy and play pivotal roles in regulating the immune response through gene regulation leading to lymphocyte activation, migration, and phagocytosis [61,72]. The IFN response is a primary antiviral response of cells [73]. We found that our immunized fish that displayed high levels of protection such as VA1 and VA2 also showed significantly elevated expressions of IL-1β, IFN-γ, and TNF-α. This indicated that inflammation-related signaling pathways had been activated and are a prerequisite for resisting viral infection. CD4 cells compose the T helper and cytotoxic T lymphocyte subsets in vertebrates and are essential for immune responsiveness [74,75]. MHC I receptors are marker for self and function in antigen presentation from intracellular pathogens, and MHC II molecules present antigenic peptides to CD4+ T cells [76,77]. In our study, most of the immunized fish had increased levels of mRNAs for these 3 genes, indicating that specific cellular immunity was evoked by the vaccine.

The majority of current fish vaccines rely on adjuvants that facilitate the early onset of immunity, prolong the duration of effector responses such as antibody formation or cytotoxic T cell activity, and can render booster immunizations dispensable [78]. The adjuvant we used in this study (Montanide IMS 1312 VG) consists of water-dispersed liquid nanoparticles that gradually deliver antigen particles, which maintains immune stimulation, enhancing both cellular and humoral immunity [44,79,80,81]. We found that the Montanide adjuvant improved protection and was most likely related to increased antigen uptake. Additionally, booster vaccination significantly increased all indicators of a successful immune response including increased ELISA and neutralizing antibody titers and expression of immune response genes, which is consistent with other vaccine studies using inactivated viruses [82,83]. Booster vaccination served to provide optimal effectiveness. 

The target organs of TiLV are the liver, spleen, and kidney [14,31,84]. We also found that these organs possessed the highest viral loads and were specifically altered by our vaccine protocols. Together, these data indicated that vaccinated fish generated neutralizing antibodies that assisted in TiLV inhibition and removal via specific immunity. This also suggested that the vaccination resulted in an inhibition of viral proliferation, although further studies are needed to specifically address this point.

## 5. Conclusions

In this study, the immunological effects and protective efficacy of formaldehyde- and β-propiolactone-inactivated vaccines against TiLV in the presence and absence of the Montanide IMS 1312 VG adjuvant in tilapia was evaluated.Results showed that the β-propiolactone-inactivated TiLV vaccine coupled with the adjuvant Montanide IMS 1312 VG and booster immunizations can provide a high level of protection from virus challenge in tilapia.This vaccine induced antibody responses in the majority of tilapia, and the titers increased significantly after booster immunization. This indicates that this vaccine could be of practical significance for the control of TiLV infection in high-risk areas for TiLV epidemics. However, further studies are required to evaluate the optimal vaccination schedule and the duration of immunity in the field.

## Figures and Tables

**Figure 1 vaccines-09-00086-f001:**
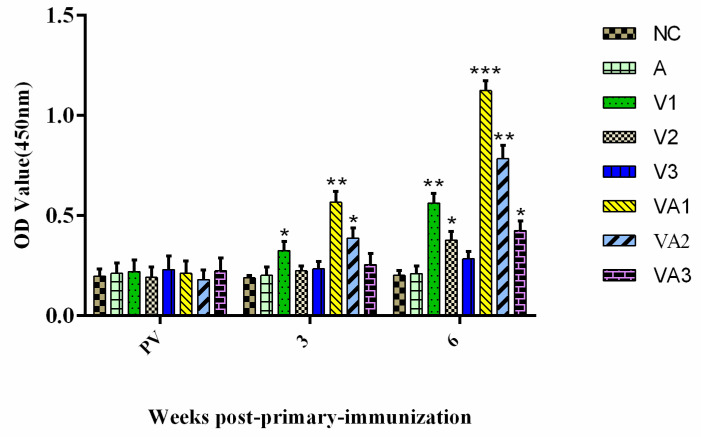
Detection of serum antibody levels induced by inactivated virus vaccine in tilapia: five blood samples of each group from the vaccinated fish were collected pre-vaccination (PV) and at three and six weeks post primary immunization (WPPI). The antibody response was determined by an indirect ELISA, and the tires were indicated at OD_450_. Data are presented as the means ± SD. A comparison between the vaccinated groups with the NC (negative control group injected with sterile PBS) or A (the adjuvant control group injected with sterile PBS mixed with the adjuvant) groups is shown. Differences in the antibody levels between vaccinated and control fish were analyzed by the Student’s *t*-test. The different numbers of asterisks indicate significant differences in antibody levels at *p* < 0.05. NC, PBS control; A, PBS and adjuvant; V1, vaccine at 1 × 10^8^ TCID_50/mL_; V2, vaccine at 1 × 10^7^ TCID_50/mL_; V3, vaccine at 1 × 10^6^ TCID_50/mL_; VA1, vaccine at 1 × 10^8^ TCID_50/mL_ and adjuvant; VA2, vaccine at 1 × 10^6^ TCID_50/mL_ and adjuvant; and VA3, vaccine at 1 × 10^6^ TCID_50/mL_ and adjuvant (see text for details).* *p* < 0.05, ** *p* < 0.01 and *** *p* < 0.001.

**Figure 2 vaccines-09-00086-f002:**
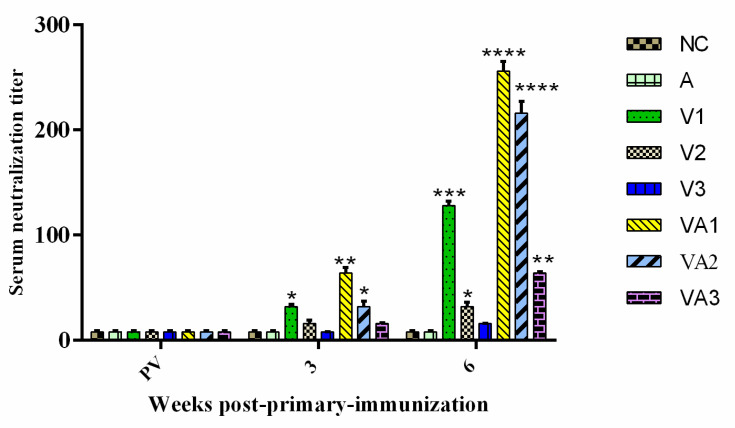
Detection of TiLV-specific serum neutralizing antibodies induced in the vaccinated tilapia: five serum samples from each group collected pre-vaccination (PV) and at three and six weeks post primary immunization (WPPI) were tested by the serum neutralization test to determine the antibody titers against the virulent isolates TiLV 2017A. A comparison between the vaccinated groups V1, V2, V3, VA1, VA2, and VA3 with the NC or A groups is shown. Differences in the antibody levels between vaccinated and control fish were analyzed by the Student’s *t*-test. The different numbers of asterisks indicate significant differences in antibody levels at * *p* < 0.05, ** *p* < 0.05, *** *p* < 0.01 and **** *p* < 0.001.

**Figure 3 vaccines-09-00086-f003:**
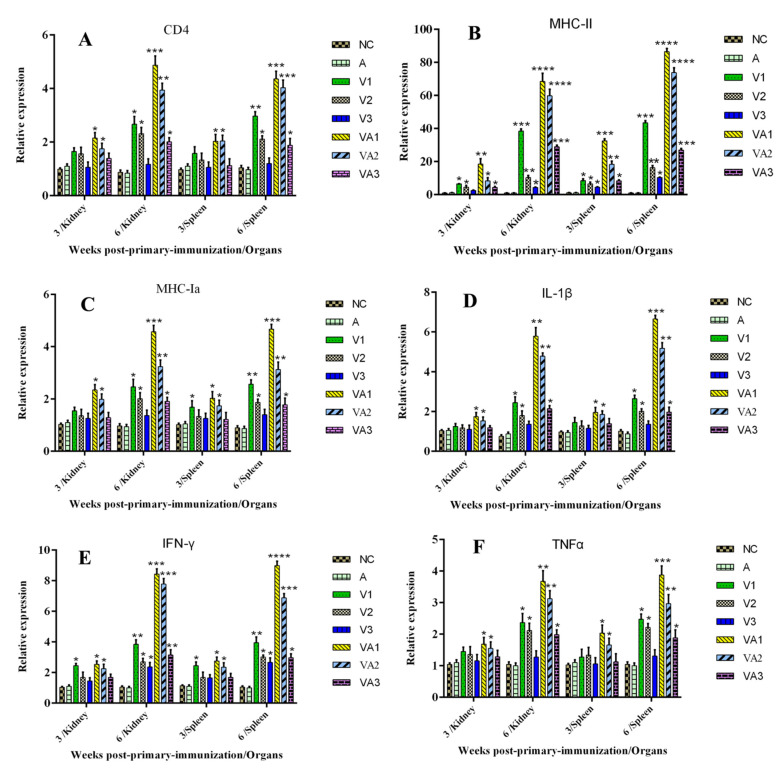
qRT-PCR analysis of the expression of immune-related genes in kidney and spleen of tilapia: (**A**) CD4, (**B**)MHC-II, (**C**) MHC-Ⅰa, (**D**) IL-1β, (**E**) IFN-γ, and (**F**) TNF-α. Five samples of each group were collected from the tilapia at three and six weeks post primary immunization (WPPI). Data were presented as the means ± SD. A comparison between the vaccinated groups V1, V2, V3, VA1, VA2, and VA3 with the NC or A groups is shown. Differences in the mRNA expression levels between vaccinated and control fish were analyzed by the Student’s *t*-test. * *p* < 0.05, ** *p* < 0.01, *** *p* < 0.001 and **** *p* < 0.0001.

**Figure 4 vaccines-09-00086-f004:**
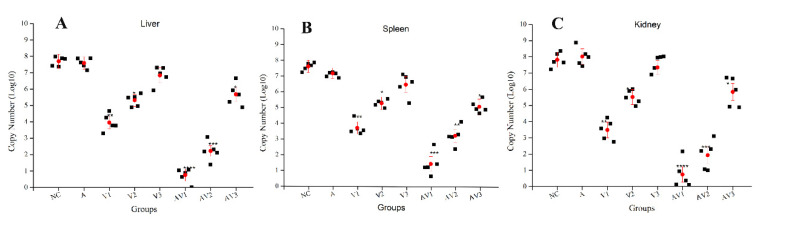
Determination of TiLV viral load from different tissues after challenge with a virulent TiLV isolate: (**A**) TiLV load in liver, (**B**) TiLV load in spleen, and (**C**) TiLV load in kidney. Data are presented as the mean ± SD. A comparison between the vaccinated groups V1, V2, V3, VA1, VA2, and VA3 with the NC or A groups is shown. The different numbers of asterisks indicate significant differences in viral load levels at *p* < 0.05.

**Figure 5 vaccines-09-00086-f005:**
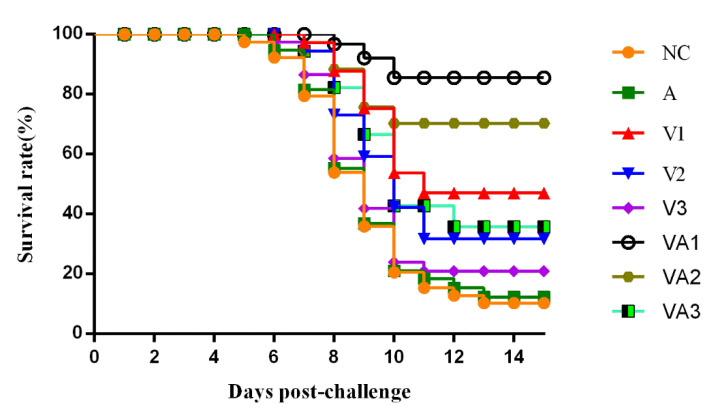
Cumulative survival rate of tilapia after challenge with 10^6^ 50% tissue culture infectious dose (TCID_50_) TiLV: The survival rate of fish was analyzed using the Kaplan–Meier survival curve using the log-rank test. One representative of three similar independent experiments is shown (*n* = 30). See Figure 1 for the experimental group abbreviations.

**Table 1 vaccines-09-00086-t001:** Infectivity and immunogenicity of tilapia lake virus (TiLV) inactivated with formaldehyde at 37 °C and β-propiolactone (BPL) at 4 °C.

Inactivators	Groups	Inactivation Temperature (°C)	Concentration	Inactivation Time (h)	Infectivity	Protection Rate (%)
Formaldehyde	1	37	1:2000	12	+	/
				24	+	/
				36	+	/
				48	+	/
				60	− *	15
	2		1:1000	12	+	/
				24	+	/
				36	− *	30
				48	−	20
				60	−	15
	3		1:500	12	+	/
				24	− *	30
				36	−	20
				48	−	20
				60	−	10
BPL	4	4	1:3000	12	+	/
				24	+	/
				36	+	/
				48	+	35
				60	− *	25
	5		1:2000	12	+	/
				24	+	/
				36	− *	45
				48	−	35
				60	−	25
	6		1:1000	12	+	/
				24	− *	35
				36	−	30
				48	−	25
				60	−	20

* Serial blind passage confirmed the absence of residual TiLV in the inactivated virus preparations. ‘+’ the virus is not completely inactivated with infectivity, ‘−’ the virus is completely inactivated without infectivity.

**Table 2 vaccines-09-00086-t002:** Genes and primer sequences used in the qRT-PCR assays.

Genes	Forward (5′-3′)	Reverse (5′-3′)	Product Size	GenBank ID	References
β-actin	GCGGAATCCACGAAACCACC	CTGTCAGCGATGCCAGGGTA	183 bp	AB270897.1	[56]
MHC-Ia	TTCTCACCAACAATGACGGG	AGGGATGATCAGGGAGAAGG	132 bp	CP003810.1	[57]
MHC-II	AGTGTGGGGAAGTTTGTTGGAT	ATGGTGACTGGAGAGAGGCG	156 bp	JN967618.1	[57]
TNFα	CTCAGAGTCTATGGGAAGCAG	GCAAACACGCCAAAGAAGGT	216 bp	NM_001279533	[57]
IFN-γ	CCAACAACTCAGGCTCGCTA	TGCTCATGGTAGCGGTGTTT	100 bp	KF294754.1	[56]
CD4	AAGAAACAGATGCGGGAGAGT	AGCAGAGGGAACGACAGAGAC	100 bp	XM_005455473.3	[58]
IL-1β	AACACTGACAGAACAACTGCGAACA	TCGCAGTTGTTCTGTCAGTGTTGTT	124 bp	XM_019365844	[57]

**Table 3 vaccines-09-00086-t003:** Mortality rate and relative percentage survival (RPS) of fish challenged with the TiLV 2017A virus.

Group	Fish No.	Mortality Rate (%)	Survival Rate (%)	RPS (%)
NC	30	93.3	6.7	/
A	30	90.0	10.0	3.5
V1	30	53.3	46.7	42.9 ^b^
V2	30	66.7	33.3	28.5 ^a^
V3	30	80.0	20.0	14.3
AV1	30	13.3	86.7	85.7 ^d^
AV2	30	33.3	66.7	64.3 ^c^
AV3	30	63.3	36.7	32.1 ^a^

Significant differences (*p* < 0.05) are indicated by different superscript letters (a, b, c and d) indicated after multiple comparisons. Differences in mortality and RPS were analyzed by one-way ANOVA with the least significant difference (LSD) post hoc test.

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
