# Peer review of "Cell Culture-Derived Tilapia Lake Virus-Inactivated Vaccine Containing Montanide Adjuvant Provides High Protection against Viral Challenge for Tilapia"

_vaccines, 2021, doi:10.3390/vaccines9020086_

Round 1
Reviewer 1 Report
The manuscript of Zeng et al. describes the primary findings about the protective effect of a beta-propiolactone-inactivated vaccine against Tilapia Lake Virus in combination with the Montanide adjuvant. The information presented here could provide a scientific basis for development of an inactivated vaccine against Tilapia Lake Virus infection in Tilapia. To improve the manuscript, could the authors consider some specific points as following?
- At line 16 of the heading 2.3 (Virus inactivation) in MM section, the inoculation volume of challenge virus should be defined.
- At line 5 of the heading 2.8 (Protection against lethal challenge...) in MM section, the meaning of “when no fish died in all groups” is unclear. In the same sentence, the authors described that the tissues were collected from “dead” fish.
- If fish serum samples were pooled for analysis of neutralizing antibodies, the authors should explain that in MM section.
- If possible, could the author add information about level of significant differences (p<0.01 or p<0.05) in Figures 1 and 2?
- In Figure 1, the IgM titer of group VA3 did not significantly increase at 3 WPPI. This is not consistent with the description in the main text at line 4 of the heading 3.2 (Detection of serum IgM and ...) in Results section.
Author Response
Dear reviewer 1
Thank you very much for giving us the opportunity to revise our submission for publication. We appreciate your thoughtful comments regarding our manuscript. We carefully took your comments and suggestions into consideration when preparing our revision and have provided updated version of the manuscript with tracked changes on the base of previous version. The responses to the review comments were point-to-point provided in part of “Responses to reviewers”.
- At line 16 of the heading 2.3 (Virus inactivation) in MM section, the inoculation volume of challenge virus should be defined.
Response: Thank you for your kind suggestion. We have defined the inoculation volume (200μL) of challenge virus in updated MS.
- At line 5 of the heading 2.8 (Protection against lethal challenge...) in MM section, the meaning of “when no fish died in all groups” is unclear. In the same sentence, the authors described that the tissues were collected from “dead” fish.
Response:Thank you for your kind comment. There were varying numbers of deaths all groups, the fish began to die on day 5 post challenge in the 2 control groups, but no fish died on days 13 to 16 post challenge in all groups. Therefore, we collected the tissues from dead fish during the experiment until 16 days post-challenge, at this time, there were no fish dead in all groups.
- If fish serum samples were pooled for analysis of neutralizing antibodies, the authors should explain that in MM section.
Response:Thank you for your kind reminder. The fish serum samples were not pooled. The ‘pooled serum samples’ in Figure.2 legends is a slip of the pen, we have corrected it in the revised version.
- If possible, could the author add information about level of significant differences (p<0.01 or p<0.05) in Figures 1 and 2?
Response:Thank you for your kind comment. The different numbers of asterisks indicate the significant differences (p<0.01 or p<0.05) in Figures 1 and 2.
- In Figure 1, the IgM titer of group VA3 did not significantly increase at 3 WPPI. This is not consistent with the description in the main text at line 4 of the heading 3.2 (Detection of serum IgM and ...) in Results section.
Response:Thank you for your kind reminder. I'm terribly sorry for my carelessness, it is incorrectly described in the main text, we have corrected it in the revised version(Line 277).

Reviewer 2 Report
Manuscript by Weiwei Zeng et. al. studied the two vaccine preparation made with formaldehyde- and β-propiolactone-inactivating treatments against TiLV in the presence and absence of Montanide IMS 1312 VG adjuvant. They found β-propiolactone inactivation provided better protection in their experimental settings. Although the study is interesting and well presented, I have following concerns.
How do authors explain a slightly more CD4 RNA in kidney than in spleen? Kidney don’t have much of immune cells.
Also authors should add flow cytometry data for these immune markers, which will be more informative in terms of cell population expressing these markers. Provided RNA data is from whole tissue so it is hard to tell which cells we are actually looking at. Given that authors see as many CD4 cells in kidney as they did in spleen, worries me about the specificity of their RNA probes.
Fig. 1 Authors needs to expand the Figure legends, and add details including number of animals in each group, number of times experiment was done and statistical analysis used. Same for other figures.
Fig.3 I suggested to increase font size or split figures into two, its hard to read text. Same for other figures, quality of poor and hard to read.
What is difference between control group A and NC?
Discussion seems more like Introduction. Authors need to revise, and discuss their results in the context of literature.
Author Response
Dear reviewer 2,
Thank you very much for giving us the opportunity to revise our submission for publication. We appreciate your thoughtful comments regarding our manuscript. We carefully took your comments and suggestions into consideration when preparing our revision and have provided updated version of the manuscript with tracked changes on the base of previous version. The responses to the review comments were point-to-point provided in part of “Responses to reviewer 2”.
How do authors explain a slightly more CD4 RNA in kidney than in spleen? Kidney don’t have much of immune cells.
Response: Thank you for your kind question. Kidney (especially the head-kidney) is the most important immune organ in fish, the head kidney of fish has dual functions similar to the central and peripheral immune organs of mammals, it's the home of immune cells( Panel and Secombes.The innate and adaptive immune system of fish. Infectious Disease in Aquaculture.2012, Pages 3-68). So it is a normal phenomenon that the kidney have much of immune cells and there is more CD4 RNA in kidney than in spleen.
Also authors should add flow cytometry data for these immune markers, which will be more informative in terms of cell population expressing these markers. Provided RNA data is from whole tissue so it is hard to tell which cells we are actually looking at. Given that authors see as many CD4 cells in kidney as they did in spleen, worries me about the specificity of their RNA probes.
Response: Thank you for your kind suggestion. Not like humans or mice or even pigs and chickens, due to lack of commercially available antibodies against most of immune markers in fish, detection of immune markers by flow cytometry has become very difficult. To my knowledge, evaluation of the cellular immunity in fish vaccine is mostly based on RNA data(Mo et al,2020;Moonika Haahr Marana,et al,2017;Fernando Carlos Ramos-Espinoza, et al.2020;Zeng, et al,2016;Ponnerassery S. Sudheesh, Kenneth D. Cain,2016;Zhang et al,2020;Hwang et al,2017;Ma et al, 2020; Zhu et al,2017)
Mo et al.Potential of naturally attenuated Streptococcus agalactiae as a live vaccine in Nile tilapia (Oreochromis niloticus).Aquaculture 518 (2020) 734774.
Zhu,et al.Effectivity of oral recombinant DNA vaccine against Streptococcus agalactiae in Nile tilapia.Developmental and Comparative Immunology 77 (2017) 77-87.
Moonika Haahr Marana,et al. Subunit vaccine candidates against Aeromonas salmonicida in rainbow trout Oncorhynchus mykiss.PLoS ONE,2017, 12(2):e0171944.
Fernando Carlos Ramos-Espinoza, et al.2020.A comparison of novel inactivation methods for production of a vaccine against Streptococcus agalactiae in Nile tilapia Oreochromis niloticus.Aquaculture,528,735484.
Zeng, et al.,Immunogenicity of a cell culture-derived inactivated vaccine against a common virulent isolate of grass carp reovirus.Fish & Shellfish Immunology. 2016.54 , 473-480.
Ponnerassery S. Sudheesh, Kenneth D. Cain.Optimization of efficacy of a live attenuated Flavobacterium psychrophilum immersion vaccine.Fish & Shellfish Immunology 56 (2016) 169-180.
Zhang et al.An effective live attenuated vaccine against Streptococcus agalactiae infection in farmed Nile tilapia (Oreochromis niloticus).Fish and Shellfish Immunology 98 (2020) 853-859.
Hwang et al.Montanide IMS 1312 VG adjuvant enhances the efficacy of immersion vaccine of inactivated viral hemorrhagic septicemia virus (VHSV) in olive flounder, Paralichthys olivaceus.Fish & Shellfish Immunology 60 (2017) 420-425.
Zhu et al,2017.Effectivity of oral recombinant DNA vaccine against Streptococcus agalactiae in Nile tilapia.Developmental and Comparative Immunology 77 (2017) 77-87.
Ma et al, 2020. Characterization of novel antigenic vaccine candidates for nile tilapia (Oreochromis niloticus) against Streptococcus agalactiae infection.Fish and Shellfish Immunology 105 (2020) 405-414.
Wang et al,2020.Cross-immunity in Nile tilapia vaccinated with Streptococcus agalactiae and Streptococcus iniae vaccines.Fish and Shellfish Immunology 97 (2020) 382-389.
Fig. 1 Authors needs to expand the Figure legends, and add details including number of animals in each group, number of times experiment was done and statistical analysis used. Same for other figures.
Response:Thank you for your kind suggestion. We have expanded the Fig.1 legends and added number of animals in each group and statistical analysis used in all Figure legends to be necessary in updated MS. The number of times experiment has described in the original Figure legends(...were collected pre-vaccination (PV) and at three and six weeks post-primary-immunization ..).
Fig.3 I suggested to increase font size or split figures into two, its hard to read text. Same for other figures, quality of poor and hard to read.
Response:Thank you for your kind suggestion. We have split the figure into two to make it more clearer in updated MS.
What is difference between control group A and NC?
Response:Thank you for your kind question. NC group only injected with sterile PBS, while group A means adjuvant control group injected with sterile PBS mixed with adjuvant Montanide IMS 1312 VG, we described it in Line 167-168 in MS.
Discussion seems more like Introduction. Authors need to revise, and discuss their results in the context of literature.
Response:Thank you for your kind suggestion. We have revised the discussion in updated MS.
